# Knowledge, attitude and management of hearing screening in children among family physicians in the Kingdom of Saudi Arabia

Ola Alqudah[1]*, Safa Alqudah[2], Ahmad M. Al-Bashaireh[3], Nouf Alharbi[4], Alia Mohammad Alqudah[5]

1 Department of Community Health, King Fahad Medical City, Riyadh, Saudi Arabia, 2 Faculty of Applied Medical Sciences, Department of Rehabilitation Sciences, Jordan University of Science and Technology, Irbid, Jordan, 3 Faculty of Nursing, Department of Primary Care Nursing, Al-Ahliyya Amman University, Amman, Jordan, 4 Department of Community Health, Second Cluster, Riyadh, Saudi Arabia, 5 Graduate studies, Al-Balqa Applied University, Amman, Jordan

☯ These authors contributed equally to this work.
* drola1980m@gmail.com

## Abstract

### Background

Early detection and management of hearing loss are important to develop ordinary speaking language and academic skills during childhood. Lack of knowledge by either parents or health care providers could hinder the process of hearing loss diagnosis, such that the intervention will be less effective. There is little evidence about the knowledge and practice of family physicians regarding hearing screening in Saudi Arabia and worldwide.

### Objectives

This study aimed to assess family physicians' knowledge, attitudes, and practices related to hearing loss in children. This in turn will help policy makers and educational institutions to establish and promote a program concerned with screening, diagnosis and intervention of paediatric hearing loss.

### Methods

A cross-sectional descriptive study enrolled 133 family physicians working at primary health centres in Saudi Arabia from March 2020 to September 2020. A self-reported questionnaire was used to assess the knowledge, attitudes, and practices of family physicians concerning hearing loss in children.

### Results

The majority of the participants were working under the umbrella of the Ministry of Health and around half of them did not screen any child for hearing loss. Despite that, 91.7% indicated the importance of neonatal hearing screening, 70.7% indicate infant candidacy for cochlear implant and only 33.1% know about the existence of the early hearing detection

**Data Availability Statement:** All relevant data are within the paper and its Supporting Information files.

**Funding:** The authors received no specific funding for this work.

**Competing interests:** The authors have declared that no competing interests exist.

and intervention (EHDI) governmental program in kingdom of Saudi Arabia (KSA). Participants were able to identify factors associated with hearing loss such as a family history of hearing loss (85.6%), meningitis (75%) and craniofacial anomalies (51.5%). The most frequent specialists for patient referrals were ear nose and throat ENT (75.2%) and audiologists (67.7%).

## Conclusion

This study shows that family physicians have good general background about the benefits of EHDI programs and the management of hearing loss in the paediatric population. However, it also indicated insufficient knowledge in other domains of hearing loss, including assessments and the presence of the EHDI governmental program in KSA. Further actions on the involvement of family physicians in the process of neonatal hearing screening, diagnosis and intervention for hearing impairment are needed.

## Introduction

Hearing is essential for the healthy mental and communication development of a child. According to the World Health Organization (WHO), there are 466 million people with hearing disabilities, and 34 million of them are children [1]. Early discovery of hearing loss can give babies a better chance of developing language, speech and communication skills [2]. It will also help them making the most of relationships with their families and explore the surrounding environment. Consequently, universal newborn hearing screening (UNHS) has become standard care in many countries. Screening is offered in the first month of life, and in most hospitals, it is done before the baby is discharged home.

There is a high prevalence of permanent hearing loss in the Saudi paediatric population, documented to be 7.7% [3,4]. A comprehensive random sample survey in Saudi Arabia of 9,540 children below the age of 15 years was taken between September 1997 to May 2000 [5]. The main objective of this study was to screen children with hearing impairment and focus on the prevalence of sensorineural hearing loss (SNHL). The results showed that 13% of screened children had hearing impairment [5].

Although newborn hearing screening (NHS) has provided opportunities for children to be identified directly after birth, the application of such a program is less implemented in developing countries, including Saudi Arabia [6]. Governmental agencies initiated national neonatal hearing screening in the kingdom in 2014. By the end of 2018, the National Neonatal Hearing Screening in Saudi Arabia covered 60% of newborns with a promise to achieve 90% implementation by the end of 2019 [7].

The knowledge, attitudes and practice (KAP) questionnaire identify what a specific group of individuals knows about certain topics and how they interact towards them. It is composed of three main domains: knowledge, attitudes and practice [8]. Under the knowledge section, set of questions are placed to evaluate the participant's information on a focused medical notion locally or worldwide [8]. The following attitude section covers queries used to inspect the current attitude, thoughts, misconceptions and confusions related to the studied term in addition to revealing any relevant misapprehension [9]. The last part of the KAP, the practice section, checks the associated therapeutic and preventive methods and procedures that either uncertainly or contributes to the investigated objectives. KAP questionnaire is considered

simple, cost-effective and beneficial in any program preparation [8]. Moreover, a large data set can be obtained from a single plot of KAP data in comparison with other qualitative scales [10].

The KAP questionnaire is extensively applied universally by scientists in different health fields for the sake of studying the knowledge, attitudes and practices of healthcare professionals [11–13], patients [14–16] and caregivers [17–19]. In audiology practice, the application of KAP has been found frequently in exploring different hearing issues including harmful noise exposure [20] and neonatal and preschool hearing screenings [21,22]. An adapted version of KAP has been used to investigate the attitude of Kuwaiti adults towards the individuals with hearing impairment [23]. Knowing more about the early management and intervention of hearing loss by health care providers in the Kingdom of Saudi Arabia is crucial for promoting neonatal hearing screening system. Consequently, it reduces the harmful impact of hearing loss on language, academic and psychosocial domains by preventing a late diagnosis. One of the challenges that is peculiar to developing countries and essential to the success of the screening program is the awareness of health care providers.

Regarding the parents of children with hearing impairment, one study shows that mothers are highly supported by the existence of national neonatal hearing screening. However, there is a necessity for preparing medical workers to enhance the awareness of families on the implication of newborn hearing screening [24]. A study was done by Almutairi et al. (2019) on 216 Saudi paediatricians, showing that 44% of them reported having high confidence and broad experience in explaining the screening guidelines to the parents of the infants [25]. Unfortunately, there has been no research exploring the up-to-date knowledge, attitudes and practices of Saudi family physicians regarding neonatal hearing screening. Family physicians have central role in counselling parents and caregivers about the need for follow-up visits in the process of auditory screening, evaluation and monitoring of communication and language skills [26]. By identifying the level of awareness among family specialists, any problem or defects in their knowledge can be addressed through regular updates and plans of action by the local professional organization. Additionally, holding scientific meetings and workshops will be also effective for accomplishing this goal.

In Saudi Arabia, all newborns should be screened for hearing impairment in the hospital before discharge. However, the Ministry of Health statistics showed that screening tests do not cover 30% of newborns [27]. As a result, there is an emphasis from the Ministry of Health that the family doctor in primary health care should check with the family if their child undergoes hearing screening in the hospital before discharge. In case the screening is not performed, the family doctor will refer the child for a hearing screening in the hospital as soon as possible.

The current research aimed to determine family physicians' knowledge and practices related to hearing loss. This is critical since family physicians are considered a direct source of knowledge for parents and the public. They are also responsible for identifying hearing loss, referring patients, and discussing the management plan with parents; therefore, they must have the latest information available related to different facts about hearing loss and have a high level of awareness and knowledge regarding hearing loss risk factors and management options. Knowing the level of physicians' knowledge will assist policymakers and educational groups in establishing and promoting hearing loss awareness.

## Methods and material

A cross-sectional descriptive design was deployed in this study. All participants were family physicians working in primary care centres in Saudi Arabia. The enrolment in the study was restricted to include only family medicine physicians without permitting other medical

specialists to participate. All family medicine physicians regardless of their experience were invited to participate in the study. A total of 300 participants were recruited as they were collected through networking and snowballing sampling. A paragraph describing the aim and significance of the study was provided, and then participants were asked to give consent and to answer the given questionnaires that explore their knowledge, attitudes, and practices (KAP) about auditory screening in children. The consent was compatible with the Helsinki Declaration of the World Medical Association and was part of a study approved by the Institutional Review Board at King Fahad Medical City, Riyadh, KSA.

## Ethical considerations

This study was approved by the Institutional Review Board at King Fahad Medical City Riyadh, KSA (IRB000010471). Before enrolling potential participants, researchers provided information about the purpose and significance of the study. Furthermore, researchers emphasised voluntary participation. Moreover, all participants will be informed about anonymity, confidentiality issues and the option of voluntary termination at any time. At the beginning of the questionnaire, the participants were asked to give their consent before moving forward in filling the questionnaire.

## Study tool and data collection

The data were collected using a self-reported questionnaire. The questionnaire was adapted from one developed by Moeller, White, and Shisler in 2006 [21], including 25 questions divided into two main sections: the first section includes 6 questions on demographic data and participants' clinical backgrounds, in particular qualification, years of experience, gender, age, and clinical setting. The respondents were also asked about the number of infants referred to them after neonatal hearing screening, as well as the approximate number of children with permanent sensorineural hearing impairment they checked in the last three years. The second section includes a set of 19 questions assessing participants' KAP regarding the aspect of audiology, involving diagnostic evaluation, hearing screening, and hearing loss management. The questions could differ in the type of answer by way of multiple choice or short answers. Additionally, items of the scale will be formed to target the aims of this study and measure the expected outcomes comprehensively. To assure that the questionnaire of this research is proportionate with predictable results, similar KAP questionnaires from the published literature were studied carefully and used as references. As a last step to validate our questionnaire, a pilot sample comprising 35 participants was used to investigate the coherence and comprehension of all questions.

SPSS version 16 was used to analyse data. All nominal and ordinal data are reported as frequencies and percentages, and numerical data were reported in terms of means and standard deviations. The data collected through the questionnaire were analysed using the independent t-test and chi-squared tests with experience in years, sex and specialty level as independent variables. Before that, the Kolmogorov-Smirnov test was applied to check the normality of the data. A comparison-based analysis was then carried out to investigate the interrelationships between knowledge and attitude capabilities among family medicine doctors towards neonatal hearing screening. A p-value of $\leq 0.05$ was considered statistically significant in all tests.

## Results

This study enrolled 133 family physicians; 47.4% of them were specialists and 52.6% were residents. The majority of participants were female (61.7%, 82/133), with an age less than 30 years old (44.4%, 59/113), and had less than five years of experience (58.8%, 78/133). Most

participants were practicing their role as family physicians under the umbrella of the Ministry of Health (82%, 109/133). Around 50% (65/113) of participants did not screen any child for hearing loss, and around 40% (53/113) did not confirm any case of childhood hearing loss during their last five years of experience. Conversely, around 51% (68/113) of participants have screened children for hearing loss during the child's visit to the well-baby clinic to receive vaccinations, and to assess his or her development, the family doctor asks the parents whether their child underwent a hearing screening test in the hospital. If not, the child is referred to the hospital for the necessary evaluation according to the protocol followed in these institutions. Around 60% of participants (80/113) were able to diagnose one or more children with hearing loss during their last five years of experience (Table 1).

Fig 1 shows the answers of the family physicians regarding factors that may associate with permanent hearing loss in children. On one hand, participants were able to identify factors associated with hearing loss to a varying degree. The highest frequency and correctly selected factors were a family history of hearing loss (85.6%, 113/133), meningitis (75%, 99/133), craniofacial anomalies (51.5%, 68/133) and congenital syphilis and mumps (42.4%, 56/113). The lowest frequent and correctly selected factors were cleft palate (27.3%, 36/133) and neonatal admission to the intensive care unit for more than 24 hours (21.2%, 28/133). On the other hand, participants claimed that frequent colds (22.7%, 30/133), congenital heart diseases (15.9%, 21/133), maternal age over 40 years old (13.6%, 18/133) and hypotonia (11.4%, 15/133) as factors associated with hearing loss in children; in fact, these factors are not associated with hearing loss.

**Table 1. Sociodemographic characteristics of the participants.**

| Variables | | Frequency | Percentage |
|---|---|---|---|
| **Qualification** | Family medicine specialist | 63 | 47.4% |
| | Family medicine resident | 70 | 52.6% |
| **Years of experience** | Less than 5 years | 78 | 58.6% |
| | 5 years- 10 year | 25 | 18.8% |
| | 11 year- 15 year | 13 | 9.8% |
| | More than 15 year | 17 | 12.8% |
| **Age** | Less than 30 year | 59 | 44.4% |
| | 30year-40 year | 46 | 34.6% |
| | 41 year-50 year | 16 | 12.0% |
| | More than 50 year | 12 | 9.0% |
| **Physician gender** | Male | 51 | 38.3% |
| | Female | 82 | 61.7% |
| | Ministerial health | 109 | 82.0% |
| | Royal medical services | 13 | 9.8% |
| | University | 7 | 5.3% |
| | Other | 4 | 3.0% |
| **The number of children was hearing screened by the participant** | 0 | 65 | 48.9% |
| | 1–100 | 55 | 41.4% |
| | 101–1000 | 4 | 3.0% |
| | More than 1000 | 9 | 6.8% |
| **Number of children with confirmed hearing loss that the respondents had during the last five years of practice** | 0 | 53 | 39.8% |
| | 1–10 | 76 | 57.1% |
| | 11–20 | 3 | 2.3% |
| | More than 20 | 1 | 0.8% |

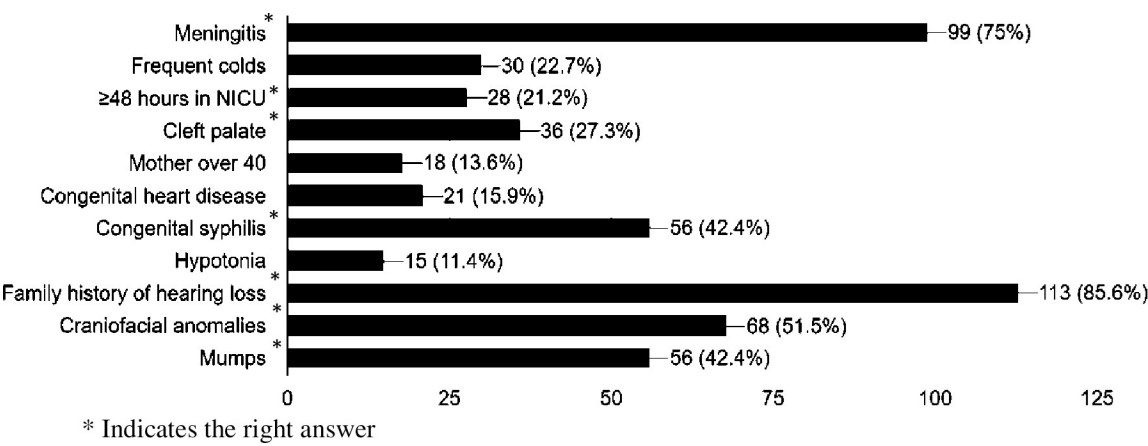

* Indicates the right answer

**Fig 1. Risk factors associated with permanent hearing loss in the pediatric population.**

Table 2 demonstrates the percentages of participants who have correctly identified the earliest age of screening for hearing, diagnosing permanent hearing loss, wearing hearing aids and starting intervention services among children. For all participants, the most frequent correct answers were for an early start to the intervention (78.9%, 105/133), followed by the earliest age for wearing a hearing aid (54.9%, 73/133). The frequency of correct answers for the remaining two items of the earliest age of screening and the diagnosis of permanent hearing loss was 41.7% (55/133) and 39.1% (52/133), respectively. Moreover, compared with the answers of the specialists, residents had a higher frequency of correct answers on all four given items.

Concerning hearing auditory screening and management, for all participants, 91.7% (122/133) indicated the importance of neonatal hearing screening, 70.7% (94/133) indicated infant candidacy for a cochlear implant and only 33.1% (44/133) knew about the existence of the EIHD governmental program in KSA (Table 3). The specialists' answers were similar to the residents' answers on these three items. Furthermore, only 40.6% (54/133) and 30.8% (41/133) were able to identify the best test to screen hearing in preschool aged children and neonates, respectively (Table 3).

Table 4 present the reasons behind the importance of early detection of hearing loss in infants. For all participants, the most frequently reported reasons were that hearing loss causes language disorders (90.2%, 119/133), poor academic achievement (76.5%, 101/133) and leads to psychological disorders (72.7%, 96/133). Table 4 provides more details about the differences in answers between the specialists and the residents.

Fig 2 shows the referral options followed by the family physician for children diagnosed with a confirmed hearing loss. The family doctor will ask parents if their child had undergone a hearing screening test at birth or if the parents noticed any symptoms in the child, and the child is referred to an audiologist for a hearing test. When the child returns for follow-up, the

**Table 2. The percentage of family physicians correctly answered the ages at which early identification of hearing loss procedures should be conducted.**

| | Overall | Specialist | Resident |
|---|---|---|---|
| **The earliest age for newborn hearing screening** | 55 (41.7%) | 24(18.2%) | 31 (23.5%) |
| **The earliest age for diagnosing permanent hearing loss in infants** | 52 (39.1%) | 20 (15.0%) | 32 (24.1%) |
| **The earliest age for wearing hearing amplification** | 73 (54.9%) | 35(26.3%) | 38 (28.6%) |
| **The earliest age for starting early intervention services** | 105 (78.9%) | 49 (36.8%) | 56 (42.1%) |

**Table 3. The percentage of family physician correctly answered the information related to hearing screening and management.**

| | Whole sample | Specialist | Resident |
|---|---|---|---|
| The importance of neonatal hearing screening | 122 (91.7%) | 61 (50%) | 61 (50%) |
| The existence of governmental program for EIHD in KSA | 44 (33.1%) | 23(52.3%) | 21 (47.3%) |
| The infant candidacy for cochlear implant | 94 (70.7%) | 48 (51.1%) | 46 (48.9%) |
| The best test to screen hearing in neonates | 41(30.8%) | 18 (43.9%) | 23 (56.1%) |
| The best test to screen hearing in preschool children | 54 (40.6%) | 26 (48.1%) | 28 (51.9%) |

reports are revised by the family doctor; if the result is abnormal, the child is then referred according to the type or cause of hearing loss to the appropriate specialty. The most frequent specialists that the family physicians referred their patients to were ENT (75.2%, 100/133), followed by the audiologists (67.7%, 90/133), (49.6%, 66/133), speech pathologists (451%, 60/133) and neurologists (15.8%, 21/133).

Table 5 demonstrates the participants' beliefs concerning the management of hearing loss in children. For all participants, the majority agreed about not prescribing hearing aids as a part of chronic otitis media treatment, disagreed that the speech therapy or auditory rehabilitation is not essential for children who have a cochlear implant and hearing aids, and disagreed that it is not essential to use amplification technology in children with one-sided deafness. Meanwhile, the majority of participants were unsure if children with a mild degree of SNHL are unfit to use hearing aids and if the auditory brainstem response (ABR) test is more accurate in estimating hearing thresholds in children than behavioural tests. Table 5 provides more details about the differences in answers between the specialists and the residents.

The extent of knowledge about EHDI procedures of both groups, i.e., residents and specialists, was not significantly different since the average knowledge score of residents (8.11± 2.01 points) was similar to that of specialists (8.37± 2.19 points; Z = −1.01, p = 0.31) based on the Mann-Whitney U test for their responses on the KAP questionnaire.

## Discussion

### Risk factors associated with permanent hearing loss

In our study, participants determined that the most frequent factors associated with permanent hearing loss in children were a family history of hearing loss, meningitis, craniofacial anomalies and congenital syphilis. To a certain extent, such findings were close to those of Campos et al. (2014) that targeted paediatricians and neonatologists and the study by Yerraguntla et al. (2016) that targeted general physicians and medical interns [28,29]. Campos et al. (2014) revealed that congenital infections, usage of ototoxic medication for more than five days, bacterial meningitis and congenital craniofacial anomalies/syndromes were the top four factors associated with permanent hearing loss in children [28]. Yerraguntla et al. (2016) reported that craniofacial anomalies, TORCH infection, trauma and a family history of hearing loss were the top four factors associated with permanent hearing loss in children [29]. It is

**Table 4. Reasons reported by family physicians explained why the early detection of hearing loss is particularly important for infants.**

| Reason | Whole sample | Specialist | Resident |
|---|---|---|---|
| Hearing loss is the most common disorder among infants | 56 (42.4%) | 32 (57.1%) | 24 (42.9%) |
| Hearing loss causes language disorders | 119 (90.2%) | 59 (49.6%) | 60 (50.4%) |
| Hearing loss causes poor academic achievement | 101 (76.5%) | 55 (54.5%) | 46 (45.5%) |
| Hearing loss leads to psychological problems | 96 (72.7%) | 47 (49%) | 49 (51%) |

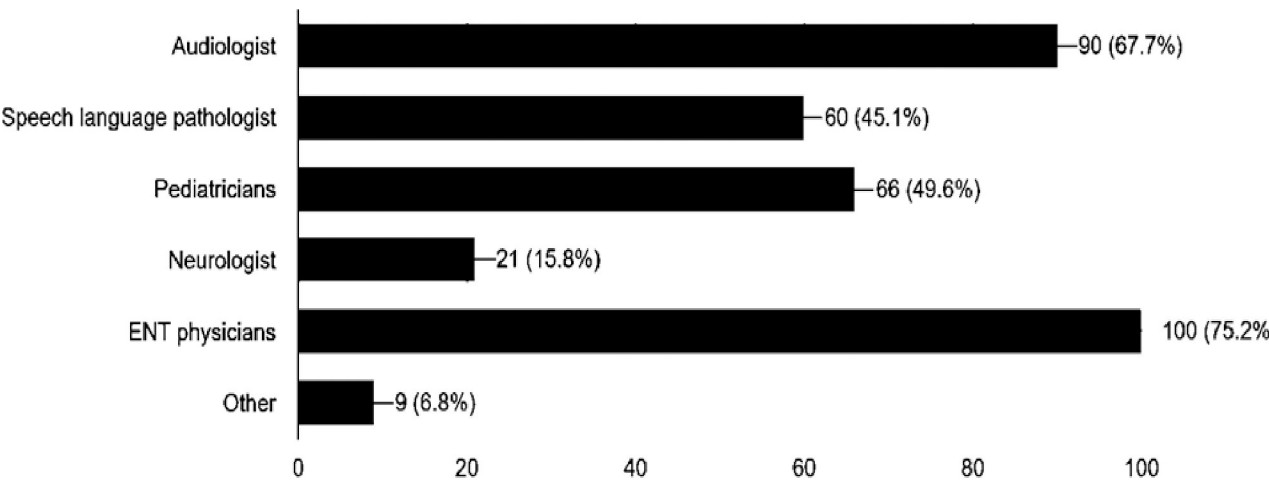

**Fig 2. The specialists that family physicians may refer to the family of a child with a confirmed permanent hearing loss.**

worth mentioning that a respectable percentage of our participants selected frequent colds, congenital heart diseases, maternal age over 40 years old and hypotonia as other potential factors that may be associated with hearing loss in children but in fact, this is not the case. Such incorrect information is expected, as a knowledge gap among various disciplines of healthcare professionals concerning the hearing loss in children has been reported in several studies [21,22,30].

## Screening and diagnosis of hearing loss in the paediatric population

Chiefly, all healthcare professionals including family physicians must be aware of the time limits concerning the screening, diagnosis and intervention of hearing loss in paediatric patients. The triad of screening, diagnosis and intervention is essential to be conducted within a specific

**Table 5. The knowledge of 133 family physicians about hearing loss management.**

| Statement | Sample | Strongly Agree | Agree | Unsure | Disagree | Strongly Disagree |
|---|---|---|---|---|---|---|
| No need to prescribe hearing aids as a part of chronic otitis media | Whole sample | 0 | 74(55.6%) | 23(17.3%) | 36 (27.1%)* | 0* |
| | Specialist | 0 | 32 (43.2%) | 13 (56.5%) | 17 (47.2%) | 0 |
| | Resident | 0 | 42 (56.8%) | 10 (43.5%) | 19 (52.8%) | 0 |
| Speech therapy and auditory rehabilitation is not essential for cochlear implanted and hearing aid children | Whole sample | 3 (2.3%) | 15 (11.3%) | 23 (17.3%) | 73 (54.9%)* | 19 (14.3%)* |
| | Specialist | 2 (66.7%) | 7 (46.7) | 13 (43.5%) | 36 (49.3%) | 10 (52.6%) |
| | Resident | 1 (33.3) | 8 (53.3%) | 10 (56.5%) | 37 (50.7%) | 9 (47.4%) |
| Children with mild degree SNHL hearing are not usually fitted with hearing aids | Whole sample | 1 (0.8%) | 29 (21.8%) | 55 (41.4.3%) | 45 (33.8%)* | 3 (2.3%)* |
| | Specialist | 0 | 14 (48.3%) | 24 (43.6%) | 23 (51.1%) | 2 (66.7%) |
| | Resident | 1 (100%) | 15 (51.7%) | 31 (56.4%) | 22 (48.9%) | 1 (33.3%) |
| It is not necessary to use hearing technology with children have one-sided deafness | Whole sample | 2 (1.5%) | 19 (14.3%) | 43 (32.4%) | 58 (43.6%)* | 11 (8.3%)* |
| | Specialist | 0 | 4 (21.1%) | 18 (41.9%) | 35 (60.3%) | 8 (72.7%) |
| | Resident | 2 (100%) | 15 (78.9%) | 25 (58.1%) | 23 (39.7%) | 3 (27.3%) |
| The ABR is more accurately estimating hearing thresholds for children than behavioral tests | Whole sample | 4 (3.0%) | 20 (15.0%) | 91 (68.4%) | 12 (9.0%)* | 6 (4.5%)* |
| | Specialist | 2 (50.0%) | 7 (35.0%) | 42 (46.2%) | 9 (75.0%) | 3 (50.0%) |
| | Resident | 2 (50.0%) | 13 (65.0%) | 49 (53.8%) | 3 (25.0%) | 3 (50.0%) |

age to maintain and restore the functioning of their hearing and speech [21,31]. Our study shows that only around 40% of our participants correctly identified the earliest age of screening (0–1 month) and diagnosis (0–3 months) for permanent hearing loss in children. Meanwhile, the participants' knowledge was better concerning the earliest age to start an intervention (78.9%) (1–6 months) and hearing aids (54.9%) (1–3 months). Campos et al. (2014) and Goedert et al (2011) reported percentages similar to our study concerning the correct age for diagnosis; however, both studies reported a higher percentage for the correct age of screening [22,28]. Interestingly, concerning the earliest age of intervention and use of hearing aids, our participants demonstrated a higher level of knowledge than that of participants in the Campos et al. (2014) and Goedert et al. (2011) studies [22,28]. A certain degree of variation in the results may be explained by the design of the reported questionnaire, the difference in the targeted population and the variation in protocols between countries.

The vast majority (91.7%) of our participants indicated the importance of neonatal hearing screening. Similar findings with a high percentage of positive attitude toward the importance of neonatal screening tests were reported in other studies [21,29,32]. Only 70.7% of our participants indicated that infants with profound bilateral hearing loss are candidates for a cochlear implant. Similar findings were reported by Mazlan and Min (2015) [33]. However, Goedert et al. (2011) reported that the participants had a great deficiency in knowledge concerning cochlear implants [22], and 40.6% and 30.8% of participants were able to determine the best screening test for preschool children (PTA) and neonates (ABR), respectively. Compared with other studies [22,28], it seems that the participants in this study had a higher level of knowledge concerning the best screening test to be used in neonates and preschool children.

## Knowledge about the importance of hearing screening programs

In 2016, screening for hearing loss was included in national screening programs aiming to reveal disorders that are quite common in the Saudi population. Since the hearing screening was recently approved by the Saudi high authorization, family medicine residents and specialists have not yet had sufficient training to improve their knowledge and clinical experience regarding hearing loss and the essence of early detection and management. Additionally, the process of implementing hearing screening programs has covered only 30 hospitals following the Ministry of Health in KSA, leading to the ineffective spread of knowledge about such programs as well as the importance of them among physicians and families. For these reasons, this current study showed that family physicians were not fully familiar with hearing screening and audiological management in KSA. Though approximately more than the half of the respondents, including family medicine residents and specialists, believed that hearing screening is required once the infant is born, they mistakenly think that hearing loss is not a common developmental disorder among infants. On the other hand, the harmful impacts of hearing loss summarized by language disorders, poor academic achievement and psychological issues were defined by most participants.

The overall review of the results indicates that family doctors are aware of textbook knowledge about neonatal screening, but they showed a lack of comprehensive understanding of the practical details of paediatric hearing impairment, such as the reasons for conducting hearing screening, specialist referral, gold standard hearing tests, hearing loss interventions and the risk factors of postnatal hearing loss. These findings were consistent with other studies that discussed knowledge regarding the early identification and management of hearing loss in children among health care providers in general [30], and specifically in primary care physicians [21,34], paediatricians [27,31,35], ENT specialists [33] and neurologists [28]. A lack of clinical application of hearing screening procedures and communication between different health

organizations and hospitals regarding early hearing detection and intervention programs show gaps in critical knowledge, warranting outreach and educational programs.

## Assessment of and interventions to treat paediatric hearing loss

Many participants were unconcerned regarding the role of hearing aids along with medication in curing unresolved conductive hearing loss in children. Despite this, clinicians frequently committed to using medications to treat conductive hearing loss, neglecting the necessity to combine antibiotics with hearing aids to treat children suffering from chronic otitis media [36]. Children always need to have intact hearing to acquire language and produce sounds precisely. With the existence of issues like recurring otitis media, a child may develop language deviance and/or learning disabilities [37,38]. Consequently, wearing a hearing aid is encouraged for children with chronic otitis media to ensure they get full access to language through listening [39].

Similarly, most of the respondents were not aware that children who wore hearing aids or cochlear implants are not capable to demonstrate speech and language competence adequately without speech therapy or hearing rehabilitation [40]. It has been scientifically proven that treating children with hearing impairment is effective at achieving (1) well-fitted hearing amplification, (2) comprehensive therapeutic programs for speech-language deviances [41].

Most family physicians were uncertain about the appropriate management when dealing with mild hearing loss in children. Additionally, they doubt the validity of behavioural measurements, such as visual reinforcement audiometry (VRA), to detect hearing impairment in the paediatric population over ABR. However, almost half of the participants agreed to consider hearing amplification as the treatment option for treating children with unilateral hearing loss. The guidelines for hearing aid fitting in children are more extended than those for adults to address unilateral and mild hearing loss patients as candidates for hearing amplification in order to ensure normal language and speech acquisition [42,43], following the American Academy of Audiology principles for paediatric amplification [44]. The influence of unilateral hearing loss or mild hearing loss impairs communication/hearing in noise (which is prevalent in the school/ kindergarten environment) as well as localization [45]; consequently, the necessity for hearing technology is required to ensure accessibility to language in children with unilateral and mild hearing loss [46].

In clinical practice, the results of this study are useful to determine the weaknesses and strengths in the understanding of family practitioners about universal hearing screening and hearing loss management. This will be extremely effective in highlighting the areas of audiology that need to be considered as educational goals for increasing the knowledge and practice levels of family physicians. For instance, the responses to the study questionnaire revealed family physicians have good knowledge of the assessment, but often fail to properly reach a diagnosis. Although family practitioners are the most in demand medical subspeciality, based on the 2019 report from a physician recruitment institution called Merritt Hawkins [47], a family clinician may rush through his or her appointments, spend less time with his patients, and ultimately not complete diagnostic procedures and refer the patient to the relevant specialists. Moreover, the family physician takes professional responsibility for managing comprehensively undifferentiated health issues and is committed to clients irrespective of their sicknesses or impacted body system. This broad scope of practice may contribute to a lack of attention focused on developing diagnostic skills for all aspects of the practice, including audiological diagnostics. Several trainings and programs in diagnosing hearing impairment, interpreting test results, intervention techniques, and parentals counselling were suggested by the study's authors and participants to improve this intention- action gap of family medicine practice in

the KSA. The limitation of this study is that the sample size was rather small, and it was not a population-based sample. Based on the information provided by the Saudi commission for health specialties, the total number of family physicians was 2,700; however, their contact information could not be obtained from representative authorities. Consequently, networking and snowballing sampling techniques were followed, leading to target 300 and recruit 133 family physicians. Future research on a larger population-based sample will be necessary to obtain more accurate and representative findings.

## Conclusion

In conclusion, this study reported that family physicians have a good general background about the benefits of EHDI programs and the management of hearing loss in the paediatric population. On the other hand, it further indicated insufficient knowledge in other domains of hearing loss including assessment, the presence of the EHDI governmental program in KSA and best therapeutic approach for persistent otitis media. The failure to acquire this information could be credited to improper implementation of EHDI programs and their limited coverage of certain urban regions in KSA. Thus, this study encourages involving family physicians in the process of neonatal hearing screening, diagnostic evaluation and intervention for hearing impairment in infants and children. As Ministry of Health organizations is responsible for delivering EHDI services, it is important that one of their actions is to reduce the harmful impact of paediatric hearing loss by preparing family practitioners to educate the families regarding the steps to ensure that their children receive adequate management if hearing loss is confirmed.

## Supporting information

**S1 File.**
(XLSX)

## Author Contributions

**Conceptualization:** Ola Alqudah.

**Data curation:** Nouf Alharbi, Alia Mohammad Alqudah.

**Formal analysis:** Safa Alqudah.

**Investigation:** Ahmad M. Al-Bashaireh.

**Methodology:** Safa Alqudah.

**Resources:** Ahmad M. Al-Bashaireh, Nouf Alharbi, Alia Mohammad Alqudah.

**Software:** Safa Alqudah, Alia Mohammad Alqudah.

**Supervision:** Ola Alqudah.

**Validation:** Ahmad M. Al-Bashaireh.

**Writing – original draft:** Ola Alqudah.

**Writing – review & editing:** Ola Alqudah, Safa Alqudah.

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
