## [Decision Letter · Decision Letter 0]

15 Feb 2021

PONE-D-21-01254

Knowledge, Attitude, and Practice of Hearing Screening and Management among Family Physicians in Kingdom of Saudi Arabia

PLOS ONE

Dear Dr. Alqudah,

Thank you for submitting your manuscript to PLOS ONE. After careful consideration, we feel that it has merit but does not fully meet PLOS ONE’s publication criteria as it currently stands. Therefore, we invite you to submit a revised version of the manuscript that addresses the points raised during the review process.

We look forward to receiving your revised manuscript.

Kind regards,

Jorge Spratley, MD, PhD

Academic Editor

PLOS ONE

Journal Requirements:

2. Please include additional information regarding the survey or questionnaire used in the study and ensure that you have provided sufficient details that others could replicate the analyses. For instance, if you developed the survey or questionnaire as part of this study and it is not under a copyright more restrictive than CC-BY, please include a copy, in both the original language and English, as Supporting Information. If the questionnaire is published, please provide a citation to the (1) questionnaire and/or (2) original publication associated with the questionnaire.

3. Thank you for stating in the text of your manuscript "This study was approved by the Institutional Review Board at King Fahad Medical City Riyadh, KSA (IRB000010471). Before enrolling potential participants, researchers provided information about the purpose and significance of the study. Furthermore, researchers emphasised voluntary participation. Moreover, all participants will be informed about anonymity, confidentiality issues and the option of voluntary termination at any time. At the beginning of the questionnaire, the participants were asked to give their consent before moving forward in filling the questionnaire." Please also add this information to your ethics statement in the online submission form.

Reviewers' comments:

Reviewer's Responses to Questions

**Comments to the Author**

1. Is the manuscript technically sound, and do the data support the conclusions?

Reviewer #1: Yes

Reviewer #2: Yes

2. Has the statistical analysis been performed appropriately and rigorously? 

Reviewer #1: Yes

Reviewer #2: Yes

3. Have the authors made all data underlying the findings in their manuscript fully available?

Reviewer #1: Yes

Reviewer #2: Yes

4. Is the manuscript presented in an intelligible fashion and written in standard English?

Reviewer #1: Yes

Reviewer #2: Yes

5. Review Comments to the Author

Reviewer #1: The content of the article is relevant and the objectives were achieved. Nevertheless there are some comments and suggestions I recommend to be considered during the revision of the paper:

I would suggest to include in the title the word children the Knowledge, Attitude, and Practice of Hearing Screening (in children) and Management among Family Physicians in Kingdom of Saudi Arabia, as the investigation relates to Hearing screening in children

When speaking about language, it should be stressed we spoken language is considered (mostly in oposition to sign language or as a complement to sign language?)

35: Methods: the total number of family physicians in the KSA that could have been enroled -that eligeble- should be mentioned

75: ....it is done before the baby is discharged home

127:They are also responsible for the identification of hearing loss and referring patients and discussing with parents the management plan; How exactly are family doctors responsible for the identification of the haering loss? What is their role? do they perform diagnostic tests? are all "refer" newborns referred do the family doctors for further diagnostic tests?

139: (related to 35) How many family medicine physicians were actually invited to participate in the study, and what is the universe of family physicians in the KSA ?

185:.... Conversely, around 51% (68/113) of participants have screened children for hearing loss and around 60% (80/113) were able to diagnose one or more children with hearing loss during their last five years of experience. How did they screened: did they participate in matternity based screening? and how did they diagnose hearing loss? do they performe diagnostic ABR or other audiological tests?

209 : Concernig #CConcerning

Figure 2 demonstrates the

221: referral options followed by the family physician for children diagnosed with a confirmed hearing loss. Who confirmed the hearing loss? The flowchart of the various stages from screening to intervention proposed in the KSA program should be explained. It seems that after the established diagnose the child goes back to the family doctor and is then referred for audiological intervention?

321: the role of surgical treatment in chronic otitis media in children should also be mentioned

322-324: this statment is a double negative. Does it mean that children with choclear implant or hearing aids do well without speech theray or hearing rehabilition ?

335:unilateral hearing loss or mild hearing loss impairs communication/hearing in noise (which is very prevalent in school/ kindergarten environment )as well as localization

Reviewer #2: The article is quite interesting and practical. It has the potential to help create awareness in the population of family physicians, on the importance of screening and diagnosis of hearing losss. It would be interesting to dive deeper into this question: if physicians have good knowledge of the assessment, but often fail to complete the diagnosis, why does this occur? Is it lack of time or attention? Which programmes or interventions could the authors suggest to improve this intention-action gap?

6. PLOS authors have the option to publish the peer review history of their article (what does this mean?). If published, this will include your full peer review and any attached files.

Reviewer #1: No

Reviewer #2: **Yes: **Dr Paulo Pessanha

---

## [Author Response · Author response to Decision Letter 0]

19 May 2021

please check the file (revision) attached with other documents.

Reviewer 1: I have incorporated all of your suggestions into my revision. they were helpful, thank you.

Reviewer 2: I have incorporated all of your suggestions into my revision. they were helpful, thank you.

---

## [Editor Report · Decision Letter 1]

24 May 2021

PONE-D-21-01254R1

Knowledge, Attitude, and Practice of Hearing Screening in Children and Management among Family Physicians in Kingdom of Saudi Arabia

PLOS ONE

Dear Dr. Alqudah,

Thank you for submitting your manuscript to PLOS ONE. After careful consideration, we feel that it has merit but does not fully meet PLOS ONE’s publication criteria as it currently stands. Therefore, we invite you to submit a revised version of the manuscript that addresses the points raised during the review process.

We look forward to receiving your revised manuscript.

Kind regards,

Jorge Spratley, MD, PhD

Academic Editor

PLOS ONE

Journal Requirements:

Additional Editor Comments (if provided):

The revised manuscript addesses the reviewers' comments and is now tehcnically sound. However the English used on the new inserts in the paper is quite poor and contrasts with the rest of the paper. As such, the manuscript will only be considered for publication following a careful grammatical revision.

---

## [Author Response · Author response to Decision Letter 1]

15 Jun 2021

The manuscript has been rechecked and the necessary changes have been made in accordance with the reviewers’ suggestions. The responses to all comments have been prepared and attached to the application

---

## [Editor Report · Decision Letter 2]

23 Jun 2021

PONE-D-21-01254R2

Knowledge, Attitude, and Practice of Hearing Screening in Children and Management among Family Physicians in the Kingdom of Saudi Arabia

PLOS ONE

Dear Dr. Alqudah,

Thank you for submitting your manuscript to PLOS ONE. After careful consideration, we feel that it has merit but does not fully meet PLOS ONE’s publication criteria as it currently stands. Therefore, we invite you to submit a revised version of the manuscript that addresses the points raised during the review process.

We look forward to receiving your revised manuscript.

Kind regards,

Jorge Spratley, MD, PhD

Academic Editor

PLOS ONE

Journal Requirements:

Additional Editor Comments (if provided):

The title should be corrected to: Knowledge, Attitude and Management of Hearing Screening in children among Family Physicians in the Kingdom of Saudi Arabia

The sentence in Abstract Line 39 and Material and Methods line 157 : "Based on the information provided by the Saudi commission for health specialties, the total number of family physicians was 2,700; however, we could not get their contact information from representative authorities, so we followed networking and snowballing sampling techniques and were able to target 300 and recruit 133 family physicians" is redundant to the previous sentence in the same sections and should be removed. Instead, it should be rewritten and sent to the end of the manuscript as a limitation of the study (133 out of 2700 family physicians).

Correct grammar in Abstract line 27: "to develop normal spoking 28 language and academic skills during childhood".

The new sequencing of the references in the revised main text is now difficult to follow, with numbers in red and black side-to-side

---

## [Author Response · Author response to Decision Letter 2]

31 Jul 2021

I have incorporated all of your suggestions into my revision. they were helpful, thank you.

---

## [Editor Report · Decision Letter 3]

12 Aug 2021

Knowledge, Attitude and Management of Hearing Screening in children among Family Physicians in the Kingdom of Saudi Arabia

PONE-D-21-01254R3

Dear Dr. Alqudah,

We’re pleased to inform you that your manuscript has been judged scientifically suitable for publication and will be formally accepted for publication once it meets all outstanding technical requirements.

Kind regards,

Jorge Spratley, MD, PhD

Academic Editor

PLOS ONE

Additional Editor Comments (optional):

Thanks for the time and effort to correct the manuscript. Congratulations for achieving the level of publication at Plos One
---

## [Editor Report · Acceptance letter]

17 Aug 2021

PONE-D-21-01254R3 

Knowledge, Attitude and Management of Hearing Screening in children among Family Physicians in the Kingdom of Saudi Arabia 

Dear Dr. Alqudah:

I'm pleased to inform you that your manuscript has been deemed suitable for publication in PLOS ONE. Congratulations! Your manuscript is now with our production department. 

Kind regards, 

on behalf of

Professor Jorge Spratley 

Academic Editor

PLOS ONE